# STAR: A Benchmark for Situated Reasoning in Real-World Videos

**Bo Wu**
MIT-IBM Watson AI Lab

**Shoubin Yu**
Shanghai Jiao Tong University

**Zhenfang Chen**
MIT-IBM Watson AI Lab

**Joshua B. Tenenbaum**
MIT BCS, CBMM, CSAIL

**Chuang Gan**
MIT-IBM Watson AI Lab

http://star.csail.mit.edu

## Abstract

Reasoning in the real world is not divorced from situations. How to capture the present knowledge from surrounding situations and perform reasoning accordingly is crucial and challenging for machine intelligence. This paper introduces a new benchmark that evaluates the situated reasoning ability via situation abstraction and logic-grounded question answering for real-world videos, called Situated Reasoning in Real-World Videos (STAR). This benchmark is built upon the real-world videos associated with human actions or interactions, which are naturally dynamic, compositional, and logical. The dataset includes four types of questions, including interaction, sequence, prediction, and feasibility. We represent the situations in real-world videos by hyper-graphs connecting extracted atomic entities and relations (*e.g.*, actions, persons, objects, and relationships). Besides visual perception, situated reasoning also requires structured situation comprehension and logical reasoning. Questions and answers are procedurally generated. The answering logic of each question is represented by a functional program based on a situation hyper-graph. We compare various existing video reasoning models and find that they all struggle on this challenging situated reasoning task. We further propose a diagnostic neuro-symbolic model that can disentangle visual perception, situation abstraction, language understanding, and functional reasoning to understand the challenges of this benchmark.

## 1 Introduction

Reasoning about real-world situations is essential to human intelligence. In a specific situation like Figure 1, we are able to know how to act in situations quickly and make feasible decisions subconsciously. That means we are logically antecedent before the concrete act. "Situated Reasoning" aims at making us understand situations dynamically and reason with the present knowledge accordingly. Such ability is logic-centered but not isolated or divorced from the surrounding situations since cognition in the real world cannot be separated from the context [5].

In fact, such situated reasoning in the real world is very challenging to existing intelligent systems. Early studies about reasoning in actions [35, 42] provide formalism definitions and frameworks from logic formalism perspectives (*e.g.*, situation calculus, *etc*.). They formulate situations as a set of formulae and perform calculus based on the designed logic rules [31, 36]. However, creating all possible logic rules in real scenarios is impossible, limiting their practicality. Recent studies of visual reasoning on synthetic video datasets [47] demonstrate the possibilities to connect visual perception, language understanding with symbolic reasoning. It remains unclear to what extent the model performs well on these synthetic datasets can be extended to real-world situations.

35th Conference on Neural Information Processing Systems (NeurIPS 2021) Track on Datasets and Benchmarks.

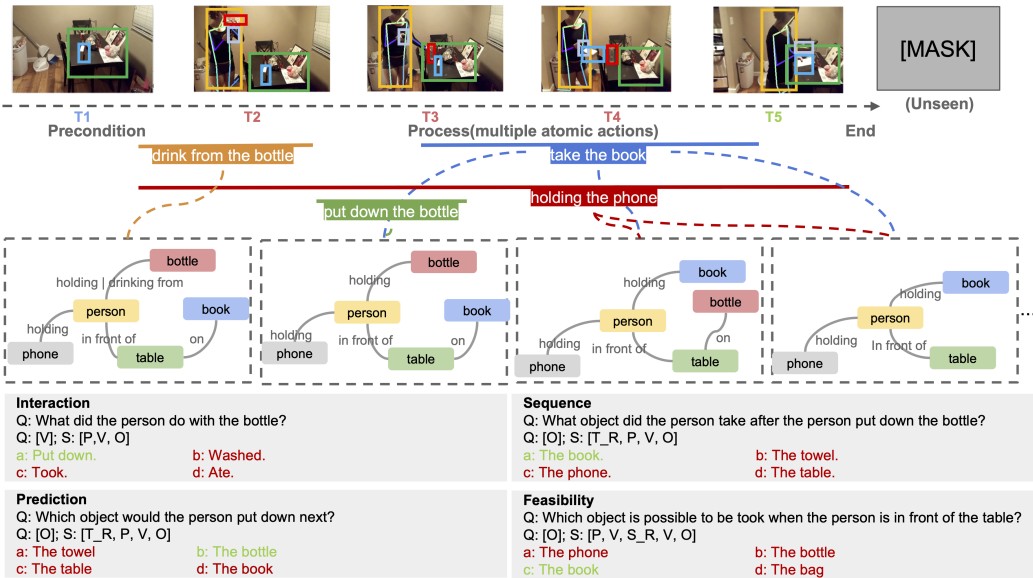

Figure 1: A representative example in the benchmark STAR. STAR aims at evaluating the skills in real-world situation recognition, abstraction, and reasoning. Q, A, and S indicate questions, answers, and situation data types with palace-holders. Answers in green (bold font) or red mean correct or incorrect. Masked situations are unseen for prediction or feasibility questions. Best viewed in color.

According to situated cognition theory [7, 5, 4], situated reasoning relies on logical thinking and integrates naturally with the present knowledge captured from the surrounding situations. Such situated reasoning may be trivial for humans but not easy to current state-of-the-art methods. According to the experiment results in Table 2 of the paper, we find existing QA models struggle with these challenging tasks, and they mainly leveraging the correlation between the visual content and question-answer pairs instead of reasoning. To explore situated reasoning with increasing complexity, we propose STAR, a novel benchmark for real-world situated reasoning via videos that require systems to capture the present knowledge from dynamic situations as structured representation and answer questions accordingly. From our perspective, such ability is a progressive process from concrete situations to mental logic. We hope the diagnostic benchmark will help to reduce the gap by conducting bottom-up perception, structured abstraction, and explicit reasoning in real-world videos.

We take human activities or actions in daily life as an exemplary domain and build the dataset upon video clips of real-world situations. The benchmark includes four types of questions: interaction question, sequence question, prediction question, and feasibility question. Each question is associated with an action-centered situation from diverse scenes and places, and each situation involves multiple actions. In order to represent the present knowledge and their dynamic changes in situations, we abstract them into structured representations with entities and relations: situation hypergraphs. Inspired by the work [22], our benchmark designs well-controlled questions and answers by question templates and programs. We simplify the language understanding by adopting concise forms and question templates for generation since our research scope mainly focuses on diagnostics for visual reasoning ability. And we also provided an auxiliary set STAR-Humans to help the evaluation with more challenging human-written questions. The answering logics describe logical reasoning processes which were grounded to executive programs over generated situation hypergraphs. We analyzed rationality by human annotations by showing these situations and synthetic questions and choices to annotators. As summarized in Table 1, STAR complements existing visual reasoning benchmarks on various aspects. It combines both situation abstraction and diagnostic reasoning focusing on human-object interaction, temporal sequence analysis, action prediction, and feasibility inference. We evaluate various visual question answering or visual reasoning models on STAR but find none of them can achieve promising performance. We design a diagnostic model called Neuro-Symbolic Situated Reasoning (NS-SR), a neural-symbolic architecture for real-world situated reasoning. It answers questions by leveraging structured situation graphs and dynamic clues from situations to perform symbolic reasoning. Our main contributions are:

- We systematically formulate the problem of situated reasoning from real-world videos, focusing on interaction, sequence, prediction, and feasibility questions.
- We construct a well-controlled benchmark STAR for situated reasoning, where designing annotations from three perspectives: visual perception, situation abstraction and logic reasoning. Each video is grounded with a situation hyper-graph, and each question is associated with a functional program that specifies the explicit reasoning steps to answer the question.
- We evaluate various state-of-the-art methods on STAR and find that they still make many mistakes in situations that are trivial for humans.
- We design a diagnostic neuro-symbolic framework for an in-depth analysis of the challenges on STAR benchmark and provide future directions on building more powerful reasoning models.

## 2 Related Work

**Visual Question Answering** Visual Question Answering [1, 40] requires a model to answer visual related questions via understanding both visual content and question semantics. The existing visual/video question answering benchmarks [13, 50, 24, 40, 44] adopted images [1, 50, 11]/videos [40, 44, 19, 28, 18, 9, 45] and types of visual comprehension questions. They achieved significant progress on evaluating the vision-language understanding ability of systems from multiple perspectives of perception. Differently, STAR requires systems to perform explicit reasoning in real-world situations and provides step-by-step reasoning programs.

**Visual Reasoning** Beyond visual question answering, several new datasets [21, 18, 47, 14, 16, 6] are designed to diagnose models' reasoning abilities. They contain questions with compositional attributes and logic programs, which require systems to perform step-by-step reasoning. It was first studied in CLEVR [21] and GQA [18] for reasoning in static images. Later, it was extended to the video domain for a more complex visual senses. MarioQA [32], COG [46], CATER [12] and CLEVRER [47] include human-annotated or generated questions and synthetic videos from simulated environments. They ask models to recognize geometric objects and their movements or collisions for understanding of compositional or spatio-temporal relations in the form of video question answering. Most of them focus on objects dynamics in synthetic scenes and it remains a doubt whether those are representative enough to reflect the complexity of real-world situations. AGQA [14] is the most recent work about reasoning in real-world videos, but it focuses on spatio-temporal relations.

**Situation Formalism** Early-stage work [31, 36, 25] establish formalisms for reasoning about action and change. The situation calculus represents changing scenarios as a set of first-order logic formulae. However, it is not realistic to apply such formalisms directly to real-world situations. Not all axioms are visible or detectable. Moreover, real-world situations are dynamic and have not been well-defined. It is still an open challenge to diagnose reasoning about actions for real-world situations.

| Dataset | Real-World Videos | Situation Abstraction | Diagnostic Reasoning | Interaction | Sequence | Prediction | Feasibility |
|---|---|---|---|---|---|---|---|
| VQA [1] | ✗ | ✗ | ✗ | ✓ | ✗ | ✗ | ✗ |
| VCR [49] | ✗ | ✗ | ✗ | ✓ | ✓ | ✗ | ✗ |
| GQA [18] | ✗ | ✓ | ✗ | ✗ | ✗ | ✗ | ✗ |
| CLEVR [21] | ✗ | ✗ | ✓ | ✗ | ✗ | ✗ | ✗ |
| COG [46] | ✗ | ✗ | ✓ | ✗ | ✗ | ✗ | ✗ |
| CLEVRER [47] | ✗ | ✗ | ✓ | ✓ | ✓ | ✓ | ✗ |
| TGIF-QA [19] | ✓ | ✗ | ✗ | ✓ | ✓ | ✗ | ✗ |
| MovieQA [40] | ✓ | ✗ | ✗ | ✓ | ✓ | ✗ | ✗ |
| TVQA/TVQA+ [28, 29] | ✓ | ✗ | ✗ | ✓ | ✓ | ✗ | ✗ |
| STAR (ours) | ✓ | ✓ | ✓ | ✓ | ✓ | ✓ | ✓ |

Table 1: Comparison between STAR and other benchmarks (visual reasoning or video QA). STAR is a real-world situated reasoning benchmark with situation abstraction and diagnostic reasoning. It contains a wide range of reasoning tasks about human-object interaction, temporal sequence analysis, action prediction, and feasibility inference.

# 3 Situated Reasoning Benchmark

STAR evaluates the human-like ability: situated reasoning. It requires systems to learn and perform reasoning in real-world situations to challenging questions. Building a situated reasoning benchmark via real-world data is challenging because it requires tight-controlled situation clues and well-designed question-answer pairs. We combine both situations abstraction and logical reasoning and adopt three guidelines in our benchmark construction: 1. situations are represented by hierarchical graphs based on bottom-up annotations for abstraction; 2. question and option generation for situated reasoning is grounded to formatted questions, functional programs, and shared situation data types; 3. situated reasoning can perform over the situation graphs iteratively.

STAR consists of about 60K situated reasoning questions with programs and answers, 240K candidate choices, and 22K trimmed situation video clips. Situation video clips in our benchmark are sourced from human activity videos, which record the dynamic interaction processes of human actions and surrounding environments in daily-life scenes. We also provide about 144K situation hypergraphs as structured situation abstraction. Designed questions cover four types of skills for situated reasoning. We constructed annotated questions with answers and options. Each question answering corresponds to a specific program for reasoning logic. To connect situation abstraction and reasoning diagnosis for question-answering, we provide situation hypergraphs tied with executable programs. Then situations, questions, and options are aligned with the unified data type schema, including actions, objects, humans, and relations. The STAR includes 111 action predicates, 28 objects, and 24 relationships. The benchmark is split into training/validation/test sets with a ratio of about 6:1:1. More dataset setting and data analysis details are in the supplementary material Section 2 or 3.

## 3.1 Situation Abstraction

**Situations** Situation is a core concept in the STAR benchmark. It describes entities, events, moments, and environments. We build up situations start from 9K source videos with action annotations sampled from Charades dataset [38]. The videos describe daily-life actions or activities in 11 indoor scenes, such as the kitchen, living room, bedroom, *etc.*. A situation is a trimmed video with multiple consecutive or overlapped actions and interactions. According to the provided annotations, we filter source videos by their quality, stability, and video length to construct clean and unambiguous data space for situations. All situation videos in our dataset are trimmed from source videos according to question types, temporal boundaries of multiple appeared actions (from Charades), and question logic. We split each action into two action segments according to the definition in situation calculus [31]: action precondition and effect. The action precondition is the beginning frame to show an initial static scene of the environment. The action effect describes the process of a single action or multiple actions. Situations of interaction or sequence questions contain complete action segments. Situations of prediction questions (or feasibility questions) include the actions involved in questions and an incomplete action effect segment (or no other action segments) about answers.

**Situation Hypergraph** To distill abstract representations from situation videos, STAR benchmark defines a unified schema to describe dynamic processes in real-world situations in the form of the hypergraph. Situation hypergraphs represent actions and inner-relations and their hierarchical structures within situations. As shown in Figure 1, each situation video is a set of subgraphs with person and object nodes, and edges represent in-frame relations (person-object or object-object). Meanwhile, each action hyperedge connects multiple subgraphs. In some cases, multiple actions are overlapped, and the nodes in subgraphs are shared. The entire dynamic process in a situation can be abstracted to a set of consecutive and overlapped situation hypergraphs. Formally, the situation hypergraph $H$ is a pair $H = (X, E)$ where $X$ is a set of nodes for objects or persons that appeared in situation frames, and $E$ is a set of non-empty hyperedge subgraphs $S_i$ for actions. Different from spatio-temporal graphs [24, 41, 20], the hypergraph structure describes actions as hyperedges and instead of the frame-level subgraphs. Such structure naturally reflects the hierarchical abstraction from real-world situations and symbolic representations. The annotations of situation hypergraphs are as follows: We created the one-to-many connections as action hyperedges based on the annotations of action temporal duration and appeared objects. The action annotations are from Charades, person-object relationships (Rel1), objects/persons annotations are from ActionGenome [20]. We extracted object-object relationships (Rel2) by using a detector VCTree with TDE [39], and extended more person-object relations (Rel3) with relation propagation over Rel1 and Rel2. For example, if <person, on, chair> and <chair, on the left of, table> exist, the <person, on the left of, table> exists. All models

in experiments use videos as inputs, but hypergraph annotations (entities, relationships, actions, or entire graphs) can be used to learn better visual perception or structured abstraction.

## 3.2 Questions and Answers Designing

The question-answer engine generates all questions, answers, and options based on situation hypergraphs. Such design allows the question and answer generation of STAR are under control and available to be applied in situated reasoning diagnosis.

**Question Generation** Situated reasoning questions ask systems to provide rational answers for multiple purposes in particular situations. We design multiple types of questions that indicate distinct purposes and cover different levels of difficulty in situated reasoning. In dynamic video situations, we propose that four types of purposes are essential and close to our daily life: happened facts, temporal order, future probability, and feasibility in a specific situation.

- **Interaction Question** (What did a person do ...): It is a basic test for understanding interactions between humans and objects in a situation.
- **Sequence Question** (What did the person do before/after ...): This type evaluates the temporal relationship reasoning of systems when facing consecutive actions in dynamic situations.
- **Prediction Question** ( What will the person do next with...): This type investigates the forecasting about plausible actions under the current situation. Seen situations only include the beginning (1/4) of actions (the remaining situations were masked), and questions ask the future actions or results.
- **Feasibility Question** (What is the person able to do/Which object is possible to be ...): This type probes the ability to infer feasible actions in particular situation conditions. We use spatial and temporal prompts (*e.g.*, spatial relationships and temporal relationships) to control the situations.

To keep the logical consistency of the question types, all types of questions are derived from well-designed templates and data from situation hypergraphs. We design formatted question templates with shared data type placeholders to align data types in situation hypergraphs (*e.g.*, [P], [O], [V], [R] for the person, objects, action verbs or relationships, *etc.*.). Then the generation process is consists of the following steps: (1) data extraction from situation annotations and hypergraphs; (2) question templates filling with extracted data; (3) language expansion for phrase collocation and morphology (articles, prepositions, and tenses).

**Answer Generation** Each question has a correct answer generated by executing a functional program (parsed from the given question) on a STAR hypergraph of a given situation video. The program shows the step-by-step reasoning process on graph structures. A valid functional program (Supplementary material Figure 5) is a set of predefined and nested functional operations that can be executed (more details refer to the work in [22]) until getting the final correct answer. Each operation takes certain entities or relationships as inputs and returns the entities, relationships, or actions as the inputs of the next reasoning step or the final output.

**Distractor Generation** Setting deliberate confusion forces systems to distinguish the reasoning logic behind correct answers and incorrect options instead of guessing by probability. We design three distractor strategies: compositional option, random option, and frequent option.

- **Compositional Option:** This option is the most challenging incorrect option since it has contraries to the given situation. It satisfies the verb-object compositionality and is also generated from the program over happened facts in the same situation.
- **Random Option:** This option also satisfies compositionality but was randomly selected from other situation hypergraphs.
- **Frequent Option:** This option is used for deceiving models by probability. It selects the most happened option in each type of question group.

Finally, all options (one correct answer and three distractors) are randomly ordered for each question.

**Debiasing and Balancing Strategies** In real-world situations, the data of human actions naturally have distribution bias and reasoning shortcuts because some entities or action compositions (e.g., "wear clothes" or "grasp doorknob") frequently occurred. Such frequent collocation makes questions can be easily answered even without seeing the actual situations or questions. To avoid such shortcuts, we control the compositionality of appeared verbs/nouns in questions and answers and only select the verbs or nouns which has multiple compositions in our dataset world. To deal with answer distribution bias, we balance the answer distribution for each type of question through

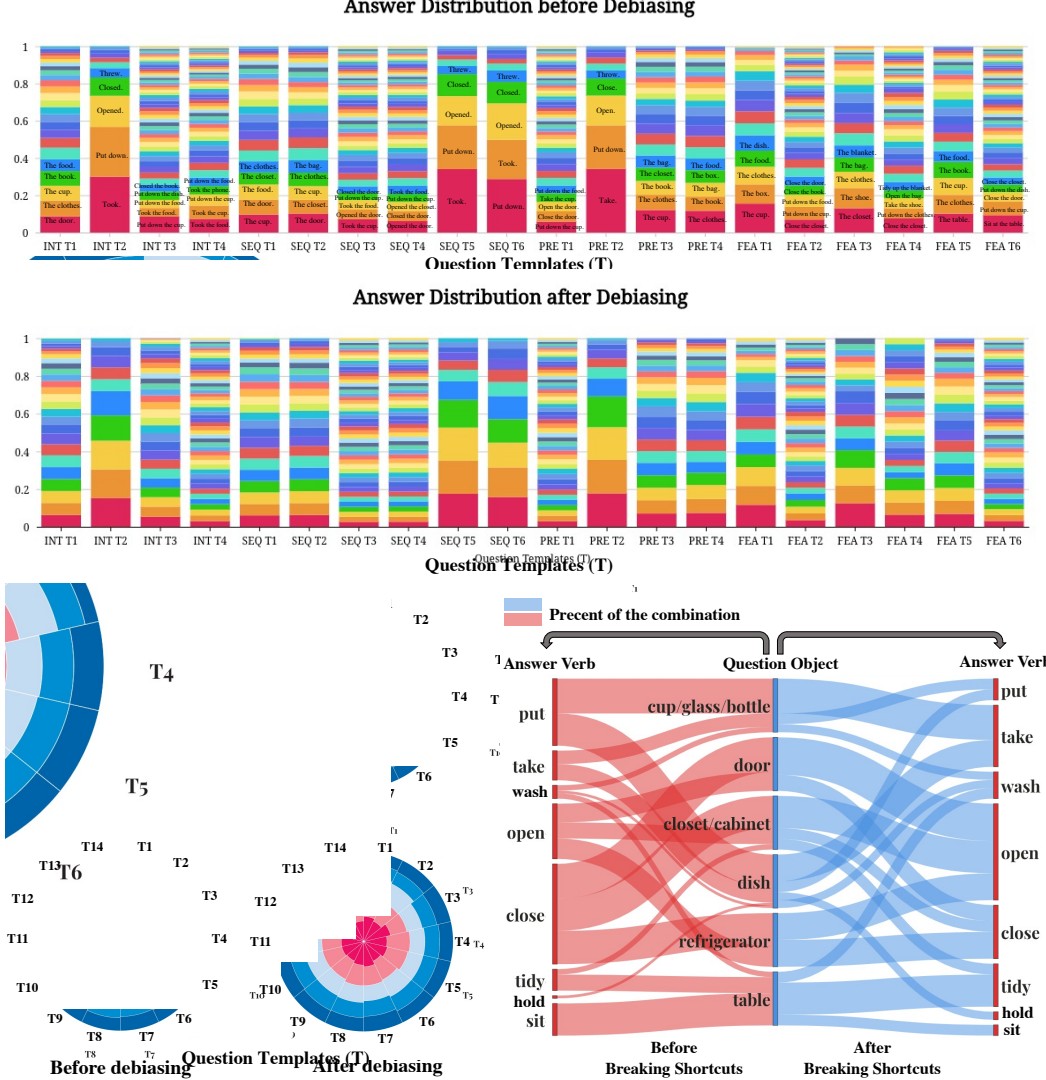

Figure 2: **Top**: The bar charts show answer distribution comparison for before and after debiasing. T: question templates; INT/SEQ/PRE/FEA: four our question types. **Bottom Left**: The bar charts show answer distribution among options, which shows STAR has a balanced distribution on each template. **Bottom Right**: The two Sankey figures illustrate the compositionality distribution change of the key components within QA pairs after breaking shortcuts. Flows mean the number of the co-occurred key components. The left subfigure shows a heavily unbalanced distribution because of the existing QA shortcuts, but the right subfigure break such shortcuts distribution after processing.

resampling. Figure 2 top and bottom right show the results of before and after the debiasing on answers and breaking shortcuts in action combinations. We notice the trend that the STAR dataset has more balanced distributions after the debiasing stage. As shown in Figure 2 bottom left, We control the frequency of entities and actions in options so that each option has a fair chance to be correct.

**Grammar Correctness and Correlation** The questions and answers in STAR are generated automatically but in the form of natural language. To validate grammar correctness, we apply grammar checkers [3, 33] to perform grammar checking and correction for word typos, tense issues, or syntactic structures. The initial grammar correctness of generated questions and answers is 87%. After three rounds of iterative corrections, the correctness achieved the expected level (improved to 98%).

**Rationality and Consistency** The real-world source videos are noisy and quality-limited because recorders captured the videos by personal phones or cameras in various indoor environments. To confirm the relevance and quality of generated situation videos, questions, and candidate choices,

we evaluate STAR through rationality and consistency by human annotation. We perform statistical analysis through a majority vote on labeled results. Three Amazon MTurk crowd-workers labeled each question. The rationality measures if a question-answer sample and the associated situation has ill-posed, semantic misaligned, or data missing issues. For each question, annotators need to label rationality by observing both questions, candidate choices, and situation videos. Here are rationality statistics in terms of four question types (from interaction to feasibility): 89.9%, 87.2%, 78.5%, and 77.5%. Consistency was calculated by the matching ratios between human-labeled options and generated options overall questions. If there is no matched correct or wrong option, this sample is none of the above makes sense. The consistency statistics of four types of questions (from interaction to feasibility) are the following: 82.5%, 85.3%, 80.4%, and 78.5%. Finally, we only keep the samples that satisfy rationality and consistency in our dataset.

## 4 Baseline Evaluation

To evaluate STAR thoroughly, we test various baseline models and analyze their strengths and weaknesses in situated reasoning. In the evaluation, a model needs to select a correct answer from the four provided candidate options for a given question. We adopt the average answer accuracy of overall questions to measure the model performance. In Table 2, we present the performances of each model individually according to the four question types. For each question, we calculate answer accuracy per question by comparing all option correctness between ground-truth and predicted results. We select representative methods for our question-answering task as competitive baselines, which include Q-type models, blind models, vision-language models, and video question-answering models. All comparison models are trained from scratch on STAR training and validation sets, and tested on the STAR test set (implementation and setting details are in the supplementary material).

| Model Name | Question Type | | | |
|---|---|---|---|---|
| | Interaction | Sequence | Prediction | Feasibility |
| Q-type (Random) [21] | 25.06 | 24.93 | 24.79 | 24.81 |
| Q-type (Frequent) [21] | 19.09 | 19.45 | 12.90 | 18.31 |
| Blind Model (LSTM) [15] | 32.24 | 32.17 | 28.56 | 28.41 |
| Blind Model (BERT) [8] | 32.68 | 34.21 | 29.98 | 29.26 |
| CNN-LSTM [47] | 33.25 | 32.67 | 30.69 | 30.43 |
| CNN-BERT [30] | 33.59 | 37.16 | 30.95 | 30.84 |
| L-GCN [17] | 39.01 | 37.97 | 28.81 | 26.98 |
| HCRN [26] | 39.10 | 38.17 | 28.75 | 27.27 |
| ClipBERT [27] | 39.81 | 43.59 | 32.34 | 31.42 |

Table 2: Question-answering accuracy results of four question types on STAR (average accuracy per question). Video QA models perform better, but significant headroom remains for further exploration.

**Q-type (Random) [21]** randomly selects a choice as answer.

**Q-type (Frequent) [21]** chooses the highest frequency answer of each question type in the train set.

**Blind Model (LSTM or BERT)** is a language-only model. We uses an LSTM [15] or transformer-based model BERT [8] to encode question and choices and a MLP to predict the answer.

**CNN+LSTM [47]** takes the final state of an LSTM to capture language and visual context.

**CNN+BERT** reimplements VL-BERT model [30] for video QA.

**L-GCN** [17] iteratively uses location-aware GCN to model object's spatial-temporal relations.

**HCRN** [26] is a recent video question answering model, which involves hierarchical conditional relation networks for better representation relation learning.

**ClipBERT** [27] is a recent state-of-the-art framework that enables end-to-end learning for video-and-language tasks including video question answering by employing sparse sampling.

### 4.1 Comparison Analysis

According to Table 2, we can conclude that STAR is a challenging task since different types of models have diverse performances, and the average level overall baselines are still not good enough.

From the results of the basic models, we can observe that the benchmark has no option biases and follows the random probability distribution naturally. The Q-type (Random) provides about 25% accuracy by randomly selecting a correct answer in four options. The Q-type (Frequent) obtain a lower performance, which indicates that the design of frequent distractors successfully influences the inference probability. With external linguistic representation as knowledge, blind models perform better than basic models only. The vision-language models can grasp the course-grained visual and language representations and achieve preliminary improvements. Nevertheless, the improvements are limited. Because simple vision-language models are good at representation but not for video question answering tasks. From simple visual-language to video QA models, about 5.03% significant increases can be observed on average accuracy. The best average accuracy achieves 36.79% by the ClipBERT. Such advantages are reasonable since they explicitly extract object interactions (LGCN) or better visual representations (HCRN and ClipBERT). We notice that although these models are better, the main improvements are from easier tasks instead of complex tasks (prediction or feasibility). These models are still struggling in reasoning tasks, although capturing vision-language interactions.

## 5    Diagnostic Model Evaluation

STAR emphasizes that ideal situation reasoning relies on visual perception, situation abstraction, and logical reasoning abilities. However, exploring the challenges and characteristics of STAR from the perspectives is not trivial. To provide more insights, we design a neuro-symbolic framework Neuro-Symbolic Situated Reasoning (NS-SR) as a diagnostic model (shown in Figure 3), which can disentangle visual perception, situation abstraction, language understanding, and symbolic reasoning. More details about implementations, evaluation, and examples are in the supplementary material.

### 5.1    Model Design

**Video Parser** This is a visual perception module consists of a set of detectors, where we obtain human-centric/object-centric interactions from video keyframe inputs. An object detector (Faster R-CNN, X101-FPN [37]) is used to detect objects/persons and RestNext-50 [43] is used to extracts visual representation for each entity. We detect relationships by VCTree with TDE-sum [39]) and extract relationship representations via GloVe [34]. A pose parser (AlphaPose [10]) is used to extract skeletons of motions. For the tasks with query actions (*e.g.*, feasibility/sequence) in questions only, we adopt a pretrained action recognizer MoViNets [23] to recognize seen actions in the situation video as preconditions. The video parser is trained on the situation video keyframes from the training set to obtain bounding box regions or visual features.

**Transformers-based Action Transition Model** To distill structured cues from the dynamic real-world situations for further reasoning, we propose a transition model to process and predict the present and future situations in the form of hypergraphs.

*Situation Hypergraph Encoder:* NS-SR performs dynamic state transitions over situation hypergraphs. The encoder constructs "initial" situation hypergraphs by connecting detected entities or relationships and encodes graphs to a structured hypergraph token sequence. Differ from existing token representations for transformers, the token sequence describes the structures of a top-down situation hypergraph and implies situation segments, subgraph segments, and entities in graphs. Suppose given $t$ situation segments $< s^0, ..., s^T >$, and each situation in time $t$ comprises multiple predicate tokens and a set of triplet tokens. Each predicate denotes an appeared atomic action $a_j$ where exists hyper-edges relation connecting a connected situation subgraph in the situation $s_t$. The triplet tokens $< h_i, o_i, r_i >$ are human-relationship-object interactions. Each situation segment is padding with zero tokens for a fixed length. We represent multiple types of embedding vectors to represent graph entities, hyper-edges, segments, and situations and sum their embeddings as a token embedding: token embedding, type or hyperedge embedding, situation embedding, position embedding, and segment embedding. The module details are in the supplementary material Section 4.

*Dynamics Transformer Model.* The dynamics model is designed to dynamically predict action states or relationships by learning the relations among the input data types in given situation videos. The model architecture is a multiple-layers of stacked transformers with down-stream task predictors. We use transformer blocks (implemented like VisualBERT [30]) to calculate self-attention scores for input token sequence with multiple heads. The attentions describe the "connections" of each potential relationship between two nodes in situation graphs (e.g., action hyper-edges or human-relationship-

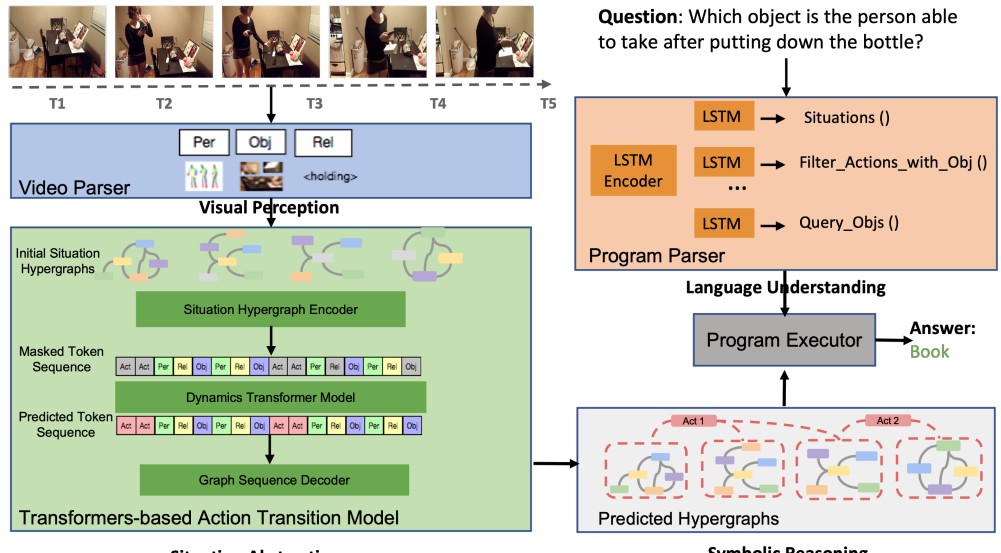

Figure 3: The architecture overview of NS-SR. It use a video parser to perceive entities, relationships and human-object interactions for visual situations. The present situation is sent to a transition model to learn complete situation abstraction and predict future situations in forms of a situation hypergraph. A program parser parses the question and options into a set of nested functions. The generated hypergraph fed to a symbolic program executor to get the answer. Best viewed in color.

object triplets *etc*..). Because the self-attention inner structures of transformers correspond with token pairs, the whole attention over input tokens performs a dynamic relation modeling. The neighbored node connections are summed into a single node. The aggregated effect will be stored in the current state in time $t$ and applied to the prediction for the missing information in the current step or the state next time $t + 1$. Such dynamic attention modeling deals with all possible relations as implicit connections. It would be more robust while relationships are unknown or some of the visual clues are not reliable. Meanwhile, we also adopt this model to predict the entities in unseen situations for prediction questions or feasibility questions.

*Graph Sequence Decoder* We set up three self-supervision tasks: action type prediction, human-object relationship type prediction, and masked token modeling (for objects or persons). The first two tasks use classifiers to predict action hyper-edges or relationships using MLPs with pooled global representations of all states in previous situations. Although recent perception models can achieve high accuracy in some datasets, some objects or human poses in our situation videos are blurred or invisible for the STAR videos. The masked token modeling aims to enhance the representation robustness by reconstructing their embedding vectors.

**Language Parser** Language Parser parses each question to a functional program [22, 47] in the form of a program sentence. The functional program (Supplementary material Figure 5 and Table 6) is composed of a series of nested operations. We defined five different types of atomic operations (*e.g.* query function) in the benchmark to construct step-by-step reasoning programs. We use an attention-based Seq2Seq model [2] to parse the input questions into corresponding programs. Since our dataset questions are single-select, we use two models to parse the questions and choices individually. Each model consists of a bidirectional LSTM encoder plus an LSTM decoder [48]. We use two hidden layers of 256 hidden units and an embedding layer to get 300-dimensional word vectors for both the encoder and decoder.

**Program Executor** We design a Program Executor to answer questions by executing programs on discrete hypergraphs (inspired by the work in [22]). It explicitly conducts the symbolic reasoning for the answering and plays the role of the reasoning engine in NS-SR. Our executor takes the program and the predicted situation hypergraph as symbolic and discrete inputs and orderly executes the mentioned functional operations in the program on the hypergraph. We implemented the predefined operations based on the entities and relations in structured situation hypergraphs (Supplementary material Table 5 and 6). Each operation inputs certain entities or relationships and outputs the predictions as the inputs of the next reasoning step or the final answer prediction. Taking hypergraphs

| NS-SR Model Variants | Question Type | | | |
|---|---|---|---|---|
| | Interaction | Sequence | Prediction | Feasibility |
| oracle version (all GT) | **100.00** | **100.00** | **100.00** | **100.00** |
| w/o perfect hypergraphs (Obj GT, Rel GT, Graph Det) | 42.61 | 46.26 | 43.44 | 43.88 |
| w/o perfect visual perception (Obj GT, Rel Det, Graph Det) | 37.47 | 38.69 | 38.49 | 38.17 |
| w/o perfect visual perception (Obj Det, Rel Det, Graph Det) | 30.89 | 31.77 | 30.24 | 29.74 |
| w/o perfect language understanding (Graph GT) | 99.97 | 99.98 | 99.98 | 99.97 |
| w/o GT | 30.88 | 31.76 | 30.23 | 29.73 |

Table 3: Performance comparison on STAR via the variants of NS-SR. GT: ground-truth, Det: detection, Obj: object, Rel: relationships, and Graph: hypergraphs.

as inputs, the reasoning starts from the cues (object, motion, or other basic data types) in questions as the initial query, then passes through all the operations iteratively and outputs the answer finally.

## 5.2 Result Analysis

Due to the modularization of NS-SR, we can explore the core challenges of STAR by an outcome-controlled evaluation under perfect/imperfect switching settings (details in the supplementary material), as shown in Table 3. Specifically, we first use all ground-truths with a symbolic reasoning module to build an oracle model, achieving the op-line accuracy (100%). This is not surprising since all questions can be answered based on perfect situation hyper-graphs and programs. Then, we remove distinct perfect conditions individually by replacing each disentangled module of NS-SR for comparisons. The final row is the performance for the version without using any ground-truths.

**Situation Abstraction:** This setting learns situation hyper-graphs by transformer-based action transition model but adopts ground-truths of the video parser (in the form of incomplete hypergraphs) and the program parser for simulation. Although having the perfect visual perception and reasoning logic, the model without perfect structured situation abstraction dropped about 55.95%. This illustrates the situation structure abstraction challenging is the bottleneck of ideal situated reasoning in STAR.

**Visual Perception:** The noticeable drops show that visual perception has a significant impact on situated reasoning. The accuracy gap between the model (using object and relationship detection) and the situation abstraction variant 13.39% is smaller than the oracle version but still significant. It denotes existing vision models struggle in real-world situations, although made remarkable progress in other tasks. And situated reasoning requires well-performed visual perception. Compared to the variants between the oracle variant, the degrades of removing relationship ground-truths larger than removing object ground-truths, which means the relationship detection has more difficulties.

**Language Understanding:** The performance without using perfect programs has slight decrease (within 1%) that implies the language perception in STAR is not difficult. It makes sense because we simplify the linguistic complexity and pays more attentions on visually-relevant reasoning challenges.

**Without Ground-Truths:** This setting uses the entire architecture in NS-SR: the video parser provides detection and poses extraction results for visual perception; the program parser provides programs parsed from given questions and options. The results are not good enough now, which shows enough remaining space for further exploration. We suggest that future directions should focus on improving the visual perception and situation abstraction on real-world videos.

## 6 Conclusions

Towards reasoning in real-world situations, we introduce a new benchmark STAR to explore how to reason accordingly. Besides perception, it integrates bottom-up situation abstraction and logical reasoning. The situation abstraction provides a unified and structured abstraction for dynamic situations, and logical reasoning adopts aligned questions, programs, and data types. We design a situated reasoning task that requires systems to learn from dynamic situations and reasonable answers for the four types of questions in specific situations: interaction, sequence, prediction, and feasibility. Our experiments demonstrate that situated reasoning is still challenging to states-of-art methods. Moreover, we design a new diagnostic model with neural-symbolic architecture to explore situated reasoning. Although the situated reasoning mechanism is not fully developed, the results show the challenges of our benchmark and indicate promising future directions. We believe STAR benchmark will open up many new opportunities for real-world situated reasoning.

## Acknowledgments and Disclosure of Funding

We thank David Cox, Jessie Rosenberg, Luke Inglis, John Cohn for their helping and advice. Thanks also to Jingwei Ji for the discussion and Steven Purdy for the legal advice. This work was supported by MIT-IBM Watson AI Lab and its member company Nexplore, ONR MURI, DARPA Machine Common Sense program, ONR (N00014-18-1-2847), and Mitsubishi Electric.

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
