# OpenReview forum: "STAR: A Benchmark for Situated Reasoning in Real-World Videos"
_NeurIPS.cc/2021/Track/Datasets_and_Benchmarks/Round2 — NeurIPS 2021 Datasets and Benchmarks Track (Round 2)_

### Official Review · Reviewer_EpDi · 2021-09-19
**An interesting idea, a very good dataset, but unclear story and model**

**Rating:** 6
**Confidence:** 4

**Strengths:**

1.  The idea of using hypergraphs to represent the knowledge from the surrounding situation is very interesting.
2.  The four types of questions that authors consider for the question answering task are fair and cover different reasoning skills.
3.  The human evaluation of this benchmark based on rationality and consistency is an important step that is included in this work.
4.  Debiasing and balancing the dataset is interesting, and it could benefit from a more detailed discussion given that the paper is submitted to the Datasets and Benchmarks track.


**Weaknesses:**


1.  The role of the *Program Executer* module in the NS-SR model is perplexing and needs to be elaborated.
2.  The authors mention *functional program* but do not elaborate on it.
3.  It is not clear whether the *hypergraphs* are part of the dataset or are the output of the video parser in the proposed NS-SR model. The hypergraphs are included in the training portion of the dataset, but they are not in the test set.
    If they are inputs to the model, then:

    1.  Why are they not included in the test set?
    2.  Figure 3 is misleading, and the hypergraphs should be drawn outside the situation abstraction box.

    If they are not inputs to the model, then:

    1.  Why are they included in the train set?
    2.  The text is misleading in lines 99-100 where the authors say STAR consists of 140.7K structured situation hypergraphs.
4.  Please clarify whether the inputs to the models are videos or situation hypergraphs or both.
5.  The two variants of NS-SR without perfect visual perception in table 3 are not clear. Since the output of the visual perception goes into the situation abstraction, and the situation abstraction is perfect, then the performance of the visual perception does not make any difference unless the authors remove the situation abstraction module for this variant which is not explicitly stated in the paper.
6. The answer generation section needs more explanation.
7.  Figure 2 is not referenced in the body of the paper. Also, it will be clearer if the authors explicitly mention that T denotes templates and A denotes answers. Moreover, I think a bar chart would have been easier to understand and look at because in the polar charts the angle is important and conveys information, but in this case, the angle represents the number of questions for each template which does not change after debiasing.
8.  In line 234, the text says figure 2, but it refers to table 2 instead.
9.  Table 3 does not explicitly mention what metric is being used for performance comparison.



**Additional Feedback:**

The validation portion of the dataset cannot be accessed at the moment.

**Clarity:**

The paper is not very well organized and a little bit difficult to read and follow. There are occasional typos, misspellings, and grammatical issues that contribute to the difficulty of the paper.


**Correctness:**

The evaluations and evaluation metrics are valid. The process of creating the benchmark also seems to be correct.
The only concern is the ablation studies and results reported in table 3. The authors claim that visual perception and situation abstraction modules are the bottlenecks of the system, and if we have perfect situation abstraction (hypergraphs) and perfect visual perception, then the system is pretty good when using imperfect language understanding (99.97%), and with perfect language understanding the system becomes perfect or oracle (100%). However, since the output of the visual perception goes into the situation abstraction module, it seems that the situation abstraction is the sole bottleneck. This can be resolved if item 5 in the weaknesses section is clarified.



**Documentation:**

This work is well documented. A link to the dataset is provided. However, the validation portion of the data set cannot be accessed at the moment.


**Ethics:**

The benchmark involves videos of human activities published in prior works, and this paper uses them. Therefore, I do not see any ethical issues immediately because of this work.


**Relation To Prior Work:**

This paper relates to the prior works very well.


**Summary And Contributions:**

This paper presents a novel benchmark for situated reasoning in the form of the question-answering task. The authors also provide a diagnostic model to showcase how challenging the different aspects of this benchmark are such as visual perception, language understanding, and situation abstraction. They conclude that visual perception and situation abstraction are the hardest aspects and require further exploration.
The dataset contains annotated videos of human activities in daily life from existing datasets.

Situations are represented by hypergraphs that connect entities and relations and can show changes in the dynamic of a situation. They call this representation *situation hypergraphs* where nodes are entities such as objects and people, and edges are in-frame relations between two objects or a person and an object such as *in_front_of*. Each action is a *hyperedge* that connects subgraphs.
The annotations of the hypergraphs are all extracted from prior works.

Situation hypergraphs are used to generate the questions and answers using a question-answer engine.
There are four types of questions in the proposed benchmark including *interaction*, *sequence*, *prediction*, and *feasibility* questions. The authors propose that these four types of questions are essential and close to human daily life.
Each question corresponds to an action in a situation and each situation involves multiple actions.
They use formatted question templates using placeholders for data types including [P]erson, [O]bject, action [V]erb, and [R]elations. The questions are then generated by extracting data types from situation annotations and hypergraphs, filling the question template with them, and a final language expansion.
Each question has a correct answer generated by a tree-structured logical program that takes hypergraphs of a situation as input. They design five program modules for this purpose including *input* module, *element* module, *filter* module, *query* module, and *logic* module.
The authors also design three distractor strategies to make the model reason logically instead of guessing. These strategies include *compositional* option, *random* option, and *frequent* option.
In the end, one correct answer and three distractors are randomly ordered for each question to make the set of candidate answers.

The authors show that existing question answering and visual reasoning models cannot perform well. They develop a diagnostic model called *Neuro-Symbolic Situated Reasoning (NS-SR)* to show the challenges of the proposed benchmark.
To evaluate the performance of models, they use *average answer accuracy* across all questions.
The baseline models the authors evaluate on STAR are the following: Q-type models (random which chooses an answer randomly or frequent which selects an answer with the highest frequency in the training set), blind models (LSTM or BERT) that use language only, vision-language models, and video question-answering models.
The best average accuracy (37.32%) is achieved by the ClipBERT model which is still low and suggests that this benchmark is challenging.
Finally, the authors perform some form of ablation study to examine which parts of the proposed NS-SR model are not good enough and make the task challenging.

---

> ### Author Response · Authors · 2021-09-27
> **Initial Review Response: Eager To Receive Your Feedback**
>
> Thank you for your review and comments. We appreciate your engagement in recognizing ideas, task design, evaluation, debiasing strategies, and dataset processing. We address your questions or concerns below:
>
> **Functional Program and Role of the Program Executor @W2 & W1.**
> - In the NS-SR method, Language Parser parses each question to a functional program (refer to the examples in Figure 5 of the supplementary and [A]) in the form of a program sentence. The functional program is composed of a series of nested operations. We defined five different types of atomic operations in the benchmark to construct step-by-step reasoning programs (Table 3 of the supplementary).
> - Program Executor is designed to answer the question by executing the program on a discrete graph (inspired by the work in [A]), which explicitly conducts the symbolic reasoning for the resulting answer. Therefore, it plays the role of the reasoning engine in NS-SR. Our executor takes the program and the predicted situation hypergraph as inputs and orderly executes the mentioned functional operations in the program on the hypergraph. We implemented the predefined operations based on the entities and relations in structured situation hypergraphs (Table 2 of the supplementary). Each operation inputs certain entities or relationships and outputs the predictions as the inputs of the next reasoning step or the final answer prediction.
> **Hypergraphs in the Test Set @W3-W4.**
> - Sorry for the confusion. We clarify that hypergraphs are not the inputs of the NS-SR model.
>   - The hypergraph annotations in the train/validation set could be used for the training/validating of hypergraph leaner. But the test set annotations won’t be released to avoid potential annotation leakage.
>   - We will modify the misleading lines 99-100: STAR also generated 140.7K situation hypergraphs as graph annotations for model training/diagnosis. The hypergraphs in train/validation sets are released but the test set won’t.
>
> - Figure 3: Sorry for the confusion. We will clarify lines 270-271: The encoder constructs “initial” situation hypergraphs by connecting detected objects and relationships. The “initial” situation hypergraph generation is a step in the Situation Abstraction box of Figure 3.
> **Model Input @W5.**
> - We clarify that the model inputs of NS-SR (w/o all GT) or baseline models are videos without hypergraphs. For the NS-SR method, the hypergraph annotations (with hyperedges, entities, relationships, or entire graphs) could be used to train the video parser or action transition model as output supervisions.
>
> **Variant Setting @W6.**
> - Sorry for the confusion, the reviewer might have misunderstood our experiment setting. The situation hypergraphs of the two variants (without perfect visual perception) in Table 3  would not be perfect since they are based on the imperfect detection results of the video parser. They are aimed at evaluating the importance of the video parser modules compared with the variant without perfect hypergraphs (w/ Obj GT, Rel GT, Graph Det).
> - We updated the variant settings (added: GT: ground-truth,  Det:detection, Graph: hypergraphs) for Table 3.
> - **Table 3:** [Updated Table](https://stardata.s3.amazonaws.com/Rebuttal/tabel3.png)
> - **Planned Table:** We are preparing a new table to clarify all variant settings in Table 3.
>
> **Answer Generation @W7.**
> - We will add more explanations in the revision:
> Each question has a correct answer generated by executing a functional program (parsed from the given question) on a STAR hypergraph of a given situation video so that the program shows the step-by-step reasoning process on graph structures. A valid functional program (refer to the examples in Figure 5 of the supplementary)  is a set of predefined and nested functional operations that can be executed (more details refer to the work in [A]) until getting the final correct answer. Each operation takes certain entities or relationships as inputs and returns the entities, relationships, or actions as the inputs of the next reasoning step or the final output.
> **Figure and Table References and Notations @W8-W10.**
> - Thanks for your corrections, we will add the reference to Figure 2 in the revision and denote the abbreviations (e.g., T: question templates) in the updated figure.
> - **Planned Plot:** We are preparing and will use a new bar chart for clear visualization.
> - We will fix the reference typo (should be Table 2 rather than Figure 2) and add the metric to Table: average accuracy per question
> **Others.**
> - We fixed the validation data accessibility issue, and it can be accessed now successfully on the homepage or the link: [STAR_val.json](https://stardata.s3.amazonaws.com/Question_Answer_SituationGraph/STAR_val.json)
> - We will correct the typos or grammatical issues and add all clarifications to the revision.
>
> **References:**
> - [A] Johnson, Justin, et al. "Inferring and executing programs for visual reasoning." ICCV. 2017.

---

> > ### Author Response · Authors · 2021-09-30
> > **[Updated] Completed Response**
> >
> > **Variant Setting @W6.**
> > - Variant Setting Table: We added a new table to clarify all variant settings of Table 3 for our diagnosis model. We will update the description in the revision of the supplementary.
> > - **Table A: Variant Settings for the Diagnosis Experiment**
> >
> > |Method Setting  |Object Detection GT  |Relationship Detection GT |Hypergraph Prediction GT |Language Parsing GT |
> > | -------- | -------- | -------- |-------- |-------- |
> > |oracle version (all GT) |✅   |✅   |✅  |✅   |
> > |w/o perfect hypergraphs  (Obj GT, Rel GT, Graph Det) |✅    |✅    |❌ predicted  |✅    |
> > |w/o perfect visual perception  (Obj GT, Rel Det) |✅    |❌ detected  |❌ predicted  |✅    |
> > |w/o perfect visual perception  (Obj Det, Rel Det) |❌ detected  |❌ detected  |❌ predicted  |✅    |
> > |w/o perfect language understanding  (Graph GT) |✅     |✅     |✅     |❌ predicted   |
> > |w/o GT  |❌ detected  |❌ detected  |❌ predicted  |❌ predicted  |
> >
> >
> > **Figure and Table References and Notations @W8-W10.**
> > - Changed to Bar Chart: We changed the figure to bar charts for clear visualization. Figure A shows the answer option/word distribution of the STAR dataset (for more details, please refer to the response for Reviewer-2 Figure A). We will update the figure in the revision
> > - **Figure A: Answer option/word distribution comparison for before and after debiasing.** [Figure A](https://stardata.s3.amazonaws.com/Rebuttal/Figure+A.pdf) (click to show the figure)
> >
> > **About the bottleneck @Correctness**
> > > The only concern is the ablation studies and results reported in table 3. The authors claim that visual perception and situation abstraction modules are the bottlenecks of the system … However, since the output of the visual perception goes into the situation abstraction module, it seems that the situation abstraction is the sole bottleneck. This can be resolved if item 5 in the weaknesses section is clarified.
> > - Sorry for the confusion. In fact, besides structured abstraction, visual perception is still an important challenge on STAR. As shown in [Table 3](https://stardata.s3.amazonaws.com/Rebuttal/tabel3.png) (comparisons: row 2 vs row 3, row 2 vs row 4), without perfect hypergraphs, the performance gaps of using perfect/imperfect perceptions are 6% - 14% in average accuracy. Meanwhile, visual perception is the basis of structured abstraction and the following steps. Following up the response for @W5, the NS-SR model (or video parser module) inputs for visual perception are videos instead of hypergraphs; action transition module (structured abstraction) inputs are built upon on video parser outputs (objects or relationships). As an upstream stage, imperfect visual perception would limit the effectiveness of structured abstraction; the detection errors would be accumulated from the perception stage to the abstraction stage.

---

> ### Author Response · Authors · 2021-09-30
> **Welcome Discussion**
>
> We wish that our response has addressed your concerns and turns your assessment to the positive side. If you have any more questions, please feel free to let us know during the rebuttal window. We appreciate your suggestions and comments!
>
> Thank you!

---

> ### Author Response · Authors · 2021-10-04
> **Look forward to your post-rebuttal feedback!**
>
> Dear Reviewer EpDi,
> Thanks for your insightful suggestions and comments again. As the deadline for discussion is approaching, we are happy to provide any additional clarifications that you may need.
>
> In our previous responses, we have carefully studied your comments and made detailed responses summarized below:
>
> - Elaborated on the role of the program executor and the implementation of the functional program (please refer to Executing Program [17] and CLEVR [16] for more details) in Section 3.2 and Section 5.1 of the paper. (@W1-W2)
>
> - Clarified that the hypergraphs are part of the dataset as graph annotations instead of model inputs. (@W3-W4)
>
> - Clarified the perception setting details of the variants of NS-SR and explained the challenge diagnosis study in Table 3 of the paper. (@W5, Correctness)
>
> - Provided detailed explanations for answer generation in Section 3.2 of the paper. (@W6)
>
> - Changed to use a bar chart instead of the polar chart and provided the correct reference/abbreviation notations for Figure 2 of the paper. (@W7-W9)
>
> - Revised the paper carefully for occasional typo/grammar issues. (@Clarity)
>
> - Pointed out the section about ethics in the supplementary and provided more details. (@Ethics)
>
> We hope that the provided additional explanations and clarifications about the details have convinced you of the merits of our submission.
>
> As all other reviewers have expressed their willingness to raise the score, we sincerely look forward to your post-rebuttal feedback. Please do not hesitate to contact us if there are other clarifications we can offer. Thanks!
>
> Best,
> Paper Authors
>
> References:
> - [16] J. Johnson, B. Hariharan, L. Van Der Maaten, L. Fei-Fei, C. Lawrence Zitnick, and R. Girshick. Clevr: A diagnostic dataset for compositional language and elementary visual reasoning.  In CVPR, 2017.
> - [17] J. Johnson, B. Hariharan, L. Van Der Maaten, J. Hoffman, L. Fei-Fei, C. Lawrence Zitnick, and R. Girshick. Inferring and executing programs for visual reasoning. In CVPR, pages 2989–2998, 2017.

---

> > ### Comment · Reviewer_EpDi · 2021-10-05
> > **Thanks for the clarifications**
> >
> > Thank you for the modifications and clarifications. Most of my concerns are addressed, but some parts are still unclear.
> >
> > In the Answer Generation subsection in lines 184-190, the authors mention that they use STAR hypergraphs of a given situation video to generate answers, but in lines 108-110, they say that the hypergraphs of the test set won't be released to prevent leakage.
> > What I understood is that the authors use hypergraphs to do the following:
> > 1. generate answers to the questions in the train/valid/test set
> > 2. train/diagnose the visual perception and situation abstraction modules only during training
> >
> > However, the authors do not use the hypergraphs in the test time.
> > This is still confusing to me because if the authors do not use the hypergraphs during the test (on the test data), then it means the results in table 3 are obtained using the training set, while the results should have been reported using the test set. If they have used the test set, then they used the hypergraphs on test data to report the results for the experiments "oracle" and "w/o perfect language understanding" which contradicts their point about annotation leakage.
> >
> > Moreover, the response from the authors resolved item 5 of weaknesses. However, the point I brought up in the Correctness section about the situation abstraction being the only bottleneck remains.
> > Since the experiment "w/o perfect hypergraphs" has low performance (given that the visual perception module is perfect), then we cannot necessarily infer that in the other two experiments "w/o perfect visual perception," the problem is the visual perception since the output of visual perception is the input of the situation abstraction. What the authors mention would have been true if the experiment "w/o perfect hypergraphs" had a higher score (in other words, the situation abstraction module was good enough).
> > I understand what the authors trying to convey, and I also note that this point does not disagree with the fact that this task is challenging. But I think it is not a completely accurate conclusion, and the other possibility is that the situation abstraction is the only bottleneck, and regardless of how good the visual perception module performs, the situation abstraction makes everything worse.

---

> > > ### Author Response · Authors · 2021-10-06
> > > **Response to Reviewer EpDi**
> > >
> > > Dear Reviewer EpDi:
> > > Thanks for your detailed feedback. We carefully read your comments and provide our response as follows:
> > >
> > > **Q1: Releasing hypergraph of the test set or not**
> > >
> > > Our dataset is designed for evaluating machine intelligence on situated reasoning from raw videos.  There are two options for the dataset release:
> > > - Option 1: Similar to the existing image reasoning datasets (e.g., GQA), we can provide the video hypergraph annotations of the train/validation set but hold out the situation hypergraph annotations in the test set for model comparison. In this setting, methods can use the ground-truth hypergraph annotation from the train/validate sets for model training and evaluate their performances on the test set by the online evaluation system.
> > > - Option 2: Releasing the full dataset with all situation hypergraphs, then the researchers could report their results based on train/test/validate sets. Researchers could either use hypergraphs annotations, raw videos, object annotations, relationship annotations to report the performances for similar studies like ours in Table 3.
> > >
> > > For Option 1,  the advantage is that models could be evaluated and compared on an online evaluation system. We can easily use a leaderboard to track the progress of this dataset under the same setting. The limitation is that researchers could not diagnose their models by using different annotations of the test set as inputs. For Option 2, the advantage is that researcher could evaluate their model performances on the arbitrary settings (using situation graph annotations or not). But it will be difficult to create a leaderboard since it is hard to monitor if the researchers used the hypergraphs data from the testing set or not.
> > >
> > > Our original plan is Option 1 (releasing the test set data without ground-truth hypergraph annotations). In such a case, the only result in Table 3 that can be used for model comparison is NS-SR (w/o GT) with an average accuracy of 30.66%. As this dataset collector or challenge organizer, we still would like to understand the potential challenges of our dataset. Thus, in Table 3, we also report results of "oracle" and "w/o perfect language understanding" using the ground-truth hypergraph annotation in the test set. We would like to emphasize again that these results are only used to explore the bottleneck of the neuro-symbolic model for situated reasoning from videos, but not for model comparisons with Table 2.
> > >
> > > But we do understand the reviewers’ concern and are also happy to switch to Option 2 if the reviewers believe that this is the best option. We are also fully prepared for Option 2. The full version of the test set (with the hypergraph annotations and correct answers) could be accessed here: [STAR_Test_option2](https://stardata.s3.amazonaws.com/Question_Answer_SituationGraph/Rebuttal/STAR_test.json)
> > >
> > >
> > > **Q2. Situation abstraction is the only bottleneck.**
> > >
> > > We do strongly agree that the only bottleneck for current neuro-symbolic models is situation abstraction.  But it remains unclear if situation abstraction is the only bottleneck for other end-to-end neural models (NN).  Since given the graph annotation GT of the test set, the performances of the NN methods (in Table A of the response for Reviewer VFTo) are much lower than the NS-SR variant w/o perfect language understanding (Graph GT) reported in Table 3. Another interesting observation is that the NN model (e.g., ClipBERT) that bypasses explicit situation abstraction and symbolic reasoning could achieve better accuracy than the neuro-symbolic model given raw videos only in testing (37.32% vs 30.66% on the average accuracy). So there is no certain answer now which method is the best solution for the situated reasoning tasks. It might be also possible to train an end-to-end model on massive videos and QA data, the model could bypass explicit situation abstraction but learn the situations from a more latent representation space, though we don't find evidence now! These observations actually make this dataset more interesting since we do not have clear answers on how to build smart AI models that could achieve human-level situated reasoning ability.  We believe that this well-bias-controlled STAR dataset could open up new opportunities for both building and evaluating video reasoning models.
> > >
> > > *We wish that our response has addressed your concerns and turns your assessment to the positive side. Please do not hesitate to contact us if there are other clarifications or experiments we can offer. Thank you very much!*
> > >
> > >
> > > Best,
> > > Paper Authors

---

> > > > ### Comment · Reviewer_EpDi · 2021-10-07
> > > > **Final comment**
> > > >
> > > > Thank you for the clarifications.
> > > >
> > > > I am willing to adjust my score if the authors add these clarifications to the paper.
> > > >
> > > > I do not have a strong preference regarding the two options for releasing the test set and the hypergraphs, but I am a little bit inclined towards the second option unless the authors clearly describe the first option in the paper and provide the link to the online evaluation platform and their plan for maintaining this platform. These details are not mentioned in the paper currently. Also, given that this is a dataset and benchmark track, I think it makes more sense to release everything.
> > > > However, that is just my opinion, and the decision is up to all reviewers and the area chair.

---

> > > > > ### Author Response · Authors · 2021-10-07
> > > > > **Response to Reviewer EpDi’s comment**
> > > > >
> > > > > Dear Reviewer EpDi:
> > > > > Thank you for your response. We will add these clarifications to our paper. We are also happy to release the full dataset.

---

### Official Review · Reviewer_VFTo · 2021-09-20

**Rating:** 6
**Confidence:** 4

**Strengths:**

The motivation for this work is well laid out and fairly clear. Reasoning is a process that is contextually relevant, an aspect that many current datasets (even in the domain of video understanding) do not sufficiently capture. The closest benchmark to this work is likely CLEVRER (Yi et al. 2020) which provides many similar challenges as STAR but for synthetic videos.

Design choices for the dataset are well detailed and explained. Authors have made initial efforts to provide a more evenly distributed answer spread.

The baseline choices along with the ablations provide sufficient information to assess the relative difficulty of the dataset, and in particular to figure out exactly which components of the dataset are most challenging.

The proposed neurosymbolic baseline NS-SR doesn't outperform fully end to end methods like ClipBERT but gives insight into failure modes of such models.

**Weaknesses:**

Given that NS-SR is used for diagnostics as an example of neuro-symbolic models in the dataset, it would be quite valuable to have the same set of ablations also perfomed on end to end methods like ClipBERT.

The very high ceiling for the ablation that has imperfect language understanding implies that language created by the QA systems is simple at best and that this dataset is useful only primarily as a vision-only dataset. This is acknowledged by the authors in the analysis though I recommend a similar disclaimer be made elsewhere like the intro.

Part of the reason for the above problem could also be the relatively small vocabulary of scenes/objects/verbs/relations, it provides very limited scope for testing ablations on the complexity of the hypergraphs etc.

I am rather unclear on the actual process of data annotation for the hypergraphs, what parts (relations for hypergraphs etc) are taken from Charades/Action Genome and what is new in STAR.


**Additional Feedback:**

-

**Clarity:**

The paper is well written overall and I was able to follow along with the main ideas throughout.


**Correctness:**

I do not see any issues with correctness beyond what's detailed in the Weaknesses section.

**Documentation:**

The benchmark site and the datasheet together provide enough documentation for this dataset to be useful for future researchers.


**Ethics:**

No ethics statement is given, it would be good to have especially as a sanity check on the underlying data biases of the videos, given that many of the videos have humans in them.


**Relation To Prior Work:**

The distinction to other VQA-style works is made sufficiently clear in Table 1.


**Summary And Contributions:**

This paper provides a benchmark, dubbed STAR, to test models' abilities to reason in situated visual environments, specifically video. The dataset contains videos and parallel hyper-graphs that contain information at the entity level about the video. A question-answering engine is then provided that is meant to generate questions and answer pairs for that particular video based on the hyper-graph. Experimental results show that existing video reasoning systems struggle on this benchmark. A new neuro-symbolic approach is also provided that performs relatively well on the benchmark.

---

> ### Author Response · Authors · 2021-09-27
> **Initial Review Response: Eager To Receive Your Feedback**
>
> Thank you for your review and comments. We appreciate your engagement in recognizing motivations, dataset design, experiments, and diagnostic method. We address your questions or concerns below:
>
> **Planned Experiment: Experiments for an end-to-end model (like ClipBERT) with perception condition-controlling @W1.**
> > it would be quite valuable to have the same set of ablations also performed on end to end methods like ClipBERT
> - We are conducting new experiments now for your suggestions and questions. We will update the results during the discussion period once it is done.
> **About the Language Difficulty, Vocabulary and the Complexity of Hypergraphs @W2-3.**
> - The reason for the high-ceiling ablation of imperfect language is that we adopt question templates to simplify QA sentences. We will add the disclaimer to the introduction section in the revision: We simplify the language understanding difficulty in STAR by adopting concise forms and question templates for generation since our research scope mainly focuses on diagnostics for visual reasoning ability.
> - Notably, as introduced in lines 98-104 of our supplementary, we also provided an auxiliary set STAR-Humans to help the evaluation with more challenging human-written questions.
> - As mentioned in lines 106-107 of the paper and Table 1 of the supplementary, STAR vocabulary contains 111 action classes, 28 objects, 24 relationships on 60.9K situations. The goal of the benchmark is to diagnose the reasoning ability integrated with graph-formed structure knowledge and nested logic, so the testing scope complexity is related to the hypergraphs. STAR derives 140.7K situation hypergraphs over 60.9K situation videos which provide enough scopes on compositionality or diversity. The “Question and Program Complexity Analysis” with Figure 2 in supplementary also show the complexity of hypergraphs support the complexity of the questions or programs.
>
> **Clarify Annotations for Hypergraphs and New Data Annotations @W4.**
> - In lines 135-140 of the paper, we introduced the annotation details: “action annotations are from Charades, person-object relationships (Rel1), objects/person annotations are from Action Genome [A]. We extracted object-object relationships (Rel2) by using a detector VCTree with TDE and extended more person-object relations (Rel3) with relation propagation over Rel1 and Rel2. For example, if <person, on, chair> and <chair, on the left of, table> exist, the <person, on the left of, table> exists.”
> - New annotations in STAR benchmark: We created the one-to-many connections as action hyperedges based on the annotations of action temporal durations and appeared objects. We extracted new object-object relationships (Rel2) and extended person-object relations (Rel3). We constructed annotated questions with answers and options, each question answering is corresponding to a certain program for visual reasoning logic.
> - We will update the above details in the revision.
>
> **Ethics**
> - The ethics statement is in the first subsection (Broader Impacts) of the supplementary. All video data (relating to persons) in our dataset are from the public dataset Charades. They prevent privacy issues by data anonymization with anonymous IDs. For details, please refer to [Charades](https://prior.allenai.org/projects/charades). We will update the above details in the revision.
>
> **References:**
> - [A] Ji, Jingwei, et al. "Action genome: Actions as compositions of spatio-temporal scene graphs." In CVPR. 2020.

---

> > ### Author Response · Authors · 2021-09-30
> > **[Updated] Completed Response**
> >
> > **Experiments for an end-to-end method with perception condition-controlling @W1.**
> > > it would be quite valuable to have the same set of ablations also performed on end to end methods like ClipBERT
> > - We conducted new experiments for your comments and will add the results in the supplementary.
> > - **Table A: Performance Comparison with Different Perception Conditions for End-to-End Method (ClipBERT [B])**
> >
> > |Method  |HyperGraph  |Obj  |Rel  |Interaction  |Sequence  |Prediction  |Feasibility  |
> > | -------- | -------- | -------- |-------- |-------- |-------- |-------- |-------- |
> > |ClipBERT  |GT  |\  |\  |46.89  |54.73  |36.24  |33.17  |
> > |ClipBERT  |\  |GT  |GT  |36.31  |38.74  |32.79  |29.54  |
> > |ClipBERT  |\  |Det  |GT  |36.14  |37.39  |32.13  |28.99  |
> > |ClipBERT  |\  |Det  |Det  |33.47  |35.83  |30.22  |27.95  |
> > - We will add the description in the revision.
> >
> > **References:**
> > - [B] Lei, Jie, et al. "Less is more: Clipbert for video-and-language learning via sparse sampling." CVPR. 2021.

---

> ### Author Response · Authors · 2021-09-30
> **Welcome Discussion**
>
> We wish that our response has addressed your concerns and turns your assessment to the positive side. If you have any more questions, please feel free to let us know during the rebuttal window. We appreciate your suggestions and comments!
>
> Thank you!

---

### Official Review · Reviewer_Smzp · 2021-09-21
**Initial Review for STAR**

**Rating:** 6
**Confidence:** 4
**Clarity:** Yes.

**Strengths:**

The paper is well written. It helps the reader to understand the method to generate the dataset.

**Weaknesses:**

I have three major concerns for the proposed dataset:

1. The contribution of the dataset is vague. After reading the paper, I am still quite confused about the main difference between the "situation reasoning" and the conventional video QA. The authors proposes 4 different types of questions but they are fairly common in video QA community. Although the authors argue that "situation" is more abstractive, I am not fully convinced that it is necessary to propose a new dataset for such a subtle argument.

2. The proposed situation reasoning is very similar to question-answering with context which is a well studied area in NLP community. It requires the agents to reason the answer with specific context in the dialog history. Even though the proposed dataset uses images as input, but the reasoning module is still working on the domain of text (implicit alignment in multi-modality models). It should not be difficult to implement an add-hoc video summarization/image caption module + conventional dialog QA to tackle this problem. Additionally, the dialog QA uses open-ended answers instead of multiple choices, which is much more difficult than 1 in 4 multiple choices. I wish the authors can make more connections to the QA community in NLP.

3. Potential data bias may exist. Given the fact that the QA format is 1 in 4 multiple choices, there are some previous studies, such as the vanilla VQA, showing that data bias exists. The author should investigate in the data bias of the answer distribution among 4 choice in the  proposed dataset.

**Additional Feedback:**

N/A

**Correctness:**

The overall method to generate the dataset looks good. The authors need to more a bit more effort the investigate into the potential data bias.

**Documentation:**

It seems to be sufficient but I don't have very careful investigation on that.

**Ethics:**

Yes.

**Relation To Prior Work:**

See weakness.

**Summary And Contributions:**

This paper introduces a new dataset STAR which focuses on "Situated Reasoning" in video domain. Situation is defined as a video clip with multiple consecutive or overlapped actions and interactions. The reasoning requires the VideoQA (suggested method) to answer the questions according to video context (situation). The paper shows a systematic way to generate the question and answer meanwhile it also provides some analysis of the question answer design and potential data bias. The proposed dataset is benchmarked with serval baselines and SOTA videoQA model.

---

> ### Author Response · Authors · 2021-09-27
> **Initial Review Response: Eager To Receive Your Feedback**
>
> Thank you for your review and comments. We appreciate your engagement in writing quality. We address your questions or concerns below:
>
> **Contribution Differences with Conventional Video QA @W1.**
> - To the best of our knowledge, all conventional real-world video QA datasets do not contain step-by-step reasoning traces on visually-grounded situated hypergraphs, so the existing VQA models mainly leveraging the correlation between the visual content and question-answer pairs.
> - Compared with conventional QA, our main contributions are 1. we built up the tight-controlled benchmark to diagnose the bottom-up situated reasoning ability, which integrates bottom-up perception, structured abstraction, and diagnostic reasoning. 2. STAR provides structured situation hypergraphs for real-world videos and well-designed questions and programs for logical reasoning. 3. we propose a diagnostic method NS-SR to evaluate and analyze the challenges in the benchmark.
> - Such situated reasoning may be trivial for humans but not easy to current VQA models. According to the experiment results in Table 2 of the paper, we find existing models struggle with these challenging tasks. Therefore, we hope the diagnostic benchmark will help to reduce the gap by conducting bottom-up visual perception, graph-structured situation abstraction, and explicit reasoning in real-world videos, inspired by the situated cognition theory.
> - More differences between situated reasoning and the conventional video QA are introduced in lines 35-41 and Table 1 (benchmark comparison) of the paper. We will clarify and update all the above descriptions in the revision.
>
> **Questions about the connections with the QA in NLP @W2.**
> - We respectfully push back the comments “the proposed situated reasoning very similar to question-answering with the context in NLP community”. Human visual perception is a procedure of detecting visible objects/relationships and visually grounding them with concepts. The visually-grounded perception is a fundamental intelligence and is also not trivial to the existing vision models (refer to the challenge analysis and experiments in response @W1, CLEVR [B], or CLEVRER [C], etc.). Therefore, ideal situated reasoning requires conducting both bottom-up perception, structured abstraction, and go beyond them to support explicit reasoning. The entire procedure is still challenging in the computer vision (CV) domain (even for multiple-choice questions).
> - Besides, the situated reasoning in CV and the QA with the context in NLP (like “Dialog QA”) have significant differences. The former mainly focuses on the situated cognition ability diagnosis from the unstructured or unseen particular situations; the latter (e.g., [A], etc.) mainly study how to learn from multiple rounds of QA conversations. The answer to the next question is most frequently either in the adjacent QA chunk.
>
> **Data Bias Analysis and Processing @W3.**
> - As introduced in lines 185-193 of the paper, we discussed the potential answer distribution bias and performed the debiasing (or resampling) and balancing strategies for a tight-controlled generation. Figure 2  of the paper shows the results after the debiasing on answers and action combinations.
> - **Planned Figure: Answer Distribution on Options.** We are preparing the answer distribution visualization to provide more investigations.
> - To control and reduce the answer bias, we intentionally generate question-answer pairs based on scripted Charades videos (via controlled collection approach: “Hollywood in Homes” [D]) and compositional question-answer generation procedure (via template-program generation approach like CLEVR [B]).
>
> **References:**
> - [A] Choi, Eunsol, et al. "Quac: Question answering in context." ACL 2018.
> - [B] J. Johnson, B. Hariharan, et al. Clevr: A diagnostic dataset for compositional language and elementary visual reasoning. In CVPR, 2017.
> - [C] K. Yi, et al. Clevrer: Collision events for video representation and reasoning. In ICLR, 2020.
> - [D] Sigurdsson, Gunnar A., et al. "Hollywood in homes: Crowdsourcing data collection for activity understanding." In ECCV, 2016.

---

> > ### Author Response · Authors · 2021-09-30
> > **[Updated] Completed Response**
> >
> > **Data Bias Analysis and Processing @W3.**
> > > Potential data bias may exist. Given the fact that the QA format is 1 in 4 multiple choices, there are some previous studies, such as the vanilla VQA, showing that data bias exists. The author should investigate in the data bias of the answer distribution among 4 choice in the proposed dataset.
> > - Answer Distribution for Data Bias Investigation.  Besides Figure 2  of the paper, we provided more visualization (e.g., the answer distribution among four choices) to investigate the data bias. Figure A (A1 or A2) compares the answer option/word distribution per question type in our balanced (after debiasing) STAR with the unbalanced (before debiasing) dataset. We notice the trend that the STAR dataset has more balanced distributions after the debiasing stage.
> > **- Figure A: Answer option/word distribution comparison for before and after debiasing.** [Figure A](https://stardata.s3.amazonaws.com/Rebuttal/Figure+A.pdf) (click to show the figure)
> > - We will update the distribution analysis in the revision.

---

> > > ### Comment · Reviewer_Smzp · 2021-10-04
> > > **Review updated**
> > >
> > > The response about the data bias clarified my concern. However, the response on the novelty of the motivation doesn't convince me therefore I would like to recommend a 6 (above threshold).

---

> ### Author Response · Authors · 2021-09-30
> **Welcome Discussion**
>
> We wish that our response has addressed your concerns and turns your assessment to the positive side. If you have any more questions, please feel free to let us know during the rebuttal window. We appreciate your suggestions and comments!
>
> Thank you!

---

### Official Review · Reviewer_4j47 · 2021-09-21
**Review for STAR**

**Rating:** 5
**Confidence:** 3
**Correctness:** See [Weaknesses].
**Clarity:** See [Strengths].

**Strengths:**

+ The paper is overall clear and well-written. The graphics are stylish and illustrative. Indeed, I find it to be an enjoyable read.
+ The authors did an excellent job distinguish their work from tons of counterparts on visual QA. By simultaneously focusing on both real-world data and abstracted & situated probs, the contribution benchmark itself is clear and undoubtedly novel, in my point of view.
+ I appreciate the extensive study on neural network models and the idea of utilizing neuro-symbolic models for diagnostic purpose. Although I do think some conclusions made by the authors are still unclear/may require more evidence (see [Weaknesses]), the experimental design is definitely a plus to their benchmark.
+ The benchmark itself is well-documented with rich details on data curation and debiasing and I find it helpful for readers to create similar dataset as well.


**Weaknesses:**

Having said those above, I do think the some of the technical details in this paper remain unclear, which raise my concerns on the claims made by the authors. I hope the following can be addressed in a rebuttal:
-In table 3, the authors showcase how can “perfect” vision and language understanding affect the final performances and therefore derive their conclusion on the bottleneck on top of it. However, it is still not clear how can the “imperfect” perception differ from the “perfect” one quantitively. It could become hard to justify the results in tab. 3 without these numbers as it may be possible that the perception models are just not well-trained.
As a side note, since there are multiple types of perception in this model, it will be necessary to present the full scores for each of them in native metric (mAP, accuracy, etc) and compare to SOTA.

-Another crucial claim is about the need of hypergraph. As I pointed out above, the authors may need to include the quantitative results on hypergraph prediction to ensure a bad-tuned abstraction learner is not the case.

-As I understand, the goal for the neural-symbolic model is to help figure out the bottleneck in this benchmark. However, what I’m not quite sure is **whether the results and claims about a neural-symbolic model can transfer to neural networks?** Indeed, these two types of learners work and react to the drawbacks in perception pipeline differently. The authors are encouraged to clarify on this. My suggestion is to include some extra experiments on NN with perfect perception, e.g. GT bboxes in VisualBERT baselines, etc.


**Additional Feedback:**

I may raise my score if the following points can be improved in a rebuttal:

-more quantitative results on the perception (vision, language, hypergraph) models.
-results on combing NN learners with perfect perception.


**Documentation:**

The authors provide detailed specification of their dataset.

**Ethics:**

The authors reused the video data from the public Charades dataset. However, it is not clear whether this data has any privacy issues.

**Relation To Prior Work:**

See [Strengths].

**Summary And Contributions:**

Paper presents STAR, a novel benchmark that checks all the marks in video question answering in terms of real-world videos, abstraction with context, delicate question design and diversity of prob. In their experiments, an extensive collection of neural-network-based reasoning models are examined while all of them struggles on the proposed benchmark. The authors future introduce a diagnostic neural-symbolic model to help figure out the true bottleneck and results suggested that improvement on both visual perception and situation abstraction are needed.

---

> ### Author Response · Authors · 2021-09-27
> **Initial Review Response: Eager To Receive Your Feedback**
>
> Thank you for your review and comments. We appreciate your engagement in recognizing the novel contributions, diagnostic method,  bias-controlling strategy, and writing quality. We address your questions or concerns below:
>
> **Quantitative results of perceptions @W1.**
> > it is still not clear how can the “imperfect” perception differ from the “perfect” one quantitively
> - To know “imperfect” perceptions quantitatively, we conducted new experiments on the STAR test set for vision, language, and hypergraph models respectively, as shown in Table A and Table B. Metric Notations: Acc: class prediction accuracy, mAP: mean average precision at IOU=0.5, R@K: recall at top K. (For each model, we use the same metrics with the reported performances for comparison)
> - For object or relationship detectors, we adopted pre-trained Faster-RCNN (X101-FPN)/ VCTree (TDE-sum) models and fine-tuned them on STAR train set annotations until the losses converged. Compare with the reported performances in scene graph SOTA [A, B], Table A shows our detectors achieve comparable or higher performances, although their detectors are trained on higher-quality (Visual Genome) or larger-scale (Action Genome, 0.47M object or 1.7M relationship instances) data. Thus, our visual perception models are well-trained.
> - **Table A: Visual Perception Comparision with [A] or [B] in Real-World Image/Video QA Datasets**
>
> | Perception Models | Our Detector Performances  |  [A] or [B] Detector Performances |
> |:-------------------------------------|-------------------------------------|-------------------------------------------:|
> | Obj Detector (Obj Det)| Faster-RCNN (X101-FPN) on STAR: mAP=25.67| [A] Faster-RCNN (X101-FPN) on Visual Genome: mAP=28.14|
> | Rel Detector (w/ Obj Det)| VCTree (TDE-sum) [E]  on STAR: R@20=29.01, R@50=30.31| [B] RelDN [D] on Action Genome: R@20=25.00, R@50=26.21|
> | Rel Detector (w/ Obj GT)| VCTree (TDE-sum) [E]  on STAR: R@20=40.34, R@50=41.07| [B] RelDN [D] on Action Genome: R@20=35.89, R@50=36.09|
>
>
> **Quantitative results of hypergraphs @W2.**
> > more quantitative results on the perception (vision, language, hypergraph) models.
> - To provide more details about perception models, we present the entire results of vision, language, hypergraph models in Table B.
> - **Table B: Module-level Quantitative Evaluation on STAR**
>
> | Modules (Settings)                           | Implementations    | Performances |
> |:---------------------------------|----------------------------------------|------------------:|
> | Obj Detector (Obj Det)    | Faster-RCNN (X101-FPN)	    | mAP=25.67
> | Rel Detector (w/ Obj Det)  | VCTree (TDE-sum) [E]   	    | R@20=29.01, R@50=30.31
> | Rel Detector (w/ Obj GT)   | VCTree (TDE-sum) [E]  	    | R@20=40.34, R@50=41.07
> | Language Parser            	 |  Seq2Seq                             | Acc=99.80
> | Action Transition Model (w/ Obj Det and Rel Det) | our transformer | action Acc=25.21, Obj Acc=30.61, Rel Acc=35.42
> | Action Transition Model (w/ Obj GT and Rel Det) | our transformer | action Acc=40.73, Obj Acc=72.74, Rel Acc=49.96
> | Action Transition Model (w/ Obj GT and Rel GT) |  our transformer | action Acc=51.88, Obj Acc=73.13, Rel Acc=66.45
> - We will update the above details to the revision.
>
> **Planned Experiment: Performances of NN Learners with Perfect Perception @W3.**
> > results on combing NN learners with perfect perception.
> - We are conducting the new experiments for the suggestion. We will update the results during the discussion period once it is done.
>
> **Ethics**
> - The public Charades dataset prevents privacy issues for their hired actors by data anonymization with anonymous IDs. For details, please refer to [Charades](https://prior.allenai.org/projects/charades).
>
> **References:**
> - [A] K. Tang, Y. Niu, J. Huang, J. Shi, and H. Zhang. Unbiased scene graph generation from biased training. In CVPR, 2020.
> - [B] J. Ji, R. Krishna, L. Fei-Fei, and J. C. Niebles. Action genome: Actions as compositions of spatio-temporal scene graphs. In CVPR, 2020.
> - [C] Jianwei Yang, Jiasen Lu, Stefan Lee, Dhruv Batra, and Devi Parikh. Graph R-CNN for scene graph generation. ECCV, 2018
> - [D] Rowan Zellers, Mark Yatskar, Sam Thomson, and Yejin Choi. Neural motifs: Scene graph parsing with global context. CVPR, 2018.
> - [E] Tang, Kaihua, et al. Learning to compose dynamic tree structures for visual contexts. CVPR. 2019.

---

> > ### Author Response · Authors · 2021-09-30
> > **[Updated] Completed Response**
> >
> > **Performances of NN Learners with Perfect Perception @W3.**
> > > results on combing NN learners with perfect perception.
> > - About whether the results and claims about a neural-symbolic model can transfer to neural networks. We conducted a new experiment Table C to show the QA results of two representative NN learners (ViusalBERT and ClipBERT). We simulate perfect visual perceptions by using object bounding box ground-truth features.
> > - The overall performances with GT are about 29% ~ 36% accuracy, and the results on the relatively hard question types (prediction/feasibility) are significantly lower than easy ones (interaction/sequence) with about 3% ~ 8% decrease. Such results illustrate the NN models (without structured abstraction and reasoning) struggle to answer the reasoning questions, especially for harder tasks.
> > - **Table C: Performances of NN Models with Perfect Perception on STAR**
> >
> > |Method|Visual Perception|Interaction|Sequence|Prediction|Feasibility|
> > | -------- | -------- | -------- |-------- |-------- |-------- |
> >   |ViusalBERT [F]  |perfect Obj bbox GT  |34.67  |35.89  |31.18  |31.53  |
> >   |ClipBERT [G]  |perfect Obj bbox GT  |36.32  |38.88  |30.73  |29.76  |
> >
> > - Implementation Details: The implementations for the variants followed as much as possible to standard VisualBERT and ClipBERT models and considered object-level/frame-level feature settings for NN learners. We use the (detected or ground-truth) object bounding box features as the visual inputs of VisualBERT and pooled (detected or ground-truth) object bounding box embedding as frame-level visual inputs for ClipBERT.
> >
> > We wish that our response has addressed your concerns. If you have any more questions, please feel free to let us know during the rebuttal window. We appreciate your suggestions and comments!
> >
> > Thank you!
> >
> > **References:**
> > - [F] Li, Liunian Harold, et al. "Visualbert: A simple and performant baseline for vision and language." arXiv preprint arXiv:1908.03557. 2019.
> > - [G] Lei, Jie, et al. "Less is more: Clipbert for video-and-language learning via sparse sampling." CVPR. 2021.

---

> > > ### Comment · Reviewer_4j47 · 2021-10-04
> > > **Thanks for the response**
> > >
> > > I have read the response from the authors. Some of my concerns are addressed. However, other reviews do raise additional issues, especially from Reviewer EpDi. Therefore, I decide to keep my initial rating.

---

> > > > ### Author Response · Authors · 2021-10-04
> > > > **About your concerns**
> > > >
> > > > Dear reviewer:
> > > > Thanks for your kindly reply.
> > > > We do think we have addressed all of your concerns from you and reviewer EpDi.
> > > > Could you please reconsider again based on the all responses?
> > > > Or could you give a detailed explanation to elaborate on your remained concerns?

---

> > > > > ### Comment · Reviewer_4j47 · 2021-10-04
> > > > > **Awaiting confirmation**
> > > > >
> > > > > Will raise the score given confirmation from EpDi.

---

### Author Response · Authors · 2021-09-27
**[Pre-Revision] Initial Response Updated**

Dear reviewers,
Thank you for your detailed feedback. We updated the responses to the reviewers' questions/suggestions by replying to each review. We will update the revision as well soon. Please let us know if you have further questions.

---

### Author Response · Authors · 2021-09-30
**[General Response] Revision Updated**

​​Thanks to the reviewers for their comments and questions. We appreciated that reviewers recognized our motivation, contributions, design choices, diagnostic method, documentation and provided constructive comments.

We have revised our paper according to all questions, suggestions, or concerns to include the following changes (highlighted in blue text):

- We have added new perception comparison experiments to show quantitative results on the well-turned perception (vision, language, hypergraph) models in paper Section 5.1 (Video Parser) and supplementary Section 4 (Quantitative Results of Perceptions). (R1)

- We have added new end-to-end NN model experiments via VisualBERT/ClipBERT to observe the effectiveness of different perception settings in paper Section 5.2 (Visual Perception) and supplementary Section 4 (NN Learners with Perfect Perception). (R1, R3)

- We have added new answer distribution visualization to further analyze the potential data bias and balanced answer distributions in paper Section 3.2 (Debiasing and Balancing Strategies) and paper Figure 2. We have highlighted our strategies to reduce data bias via balancing and shortcut breaking and the visualized answer distributions in Figure 2. (R2)

- We have discussed contribution, problem, or method differences between our dataset and conventional Video QA or NLP Dialog QA. (R2)

- We have elaborated on the module details (program executor, functional program, and answer generation) in paper Section 3.2 and Section 5.1. (R4)

- We have revised the paper carefully, including providing more language difficulty in Section 1, experiment settings in supplementary Table 5, annotation details in Section 3, implementation details in supplementary Table 2 or paper Section 5.1, other explanations, and grammar checking. (R1, R2, R3, R4)

Please don’t hesitate to let us know if you have any further comments on the manuscript or the changes.

---

### Decision · Program_Chairs · 2021-10-09

**Decision:**

Accept

**Comment:**

The paper proposes a video question answering benchmark that relies on scene hypergraphs connecting atomic entities and relations, to enable reasoning tasks given a video context. Questions and answers are procedurally generated, and cover interaction, sequence, prediction, and feasibility situations. Different models are evaluated on this benchmark showing their shortcomings around visual understanding of the scene and abstract reasoning, and a neuro-symbolic tool is proposed for diagnosing model failure.

Several concerns related to novelty, clarity of writing, resources release, and data bias were raised by reviewers and the authors actively engaged in providing explanations and adding clarifications to the manuscript. Following detailed discussions between reviewers and area chair, the reviewers increased their scores (or expressed a wish to increase them) to acknowledge the manuscript updates made by the authors. Given the final positive evaluation made by all reviewers, we conclude that this benchmark is useful, the data collection process is sound, so this is a valuable resource for the community and we recommend it for publication. However, we strongly encourage the authors to add all the clarifications promised in rebuttal and carefully proofread the manuscript for the camera-ready version.